# Bring Your Own Data!
# Self-Sensitivity Evaluation for Large Language Models

**Neel Jain*[†], Khalid Saifullah*[†]**
**Yuxin Wen[†], John Kirchenbauer[†], Manli Shu[†], Aniruddha Saha[†], Micah Goldblum[‡]**
**Jonas Geiping[†], Tom Goldstein[†]**
[†]University of Maryland, [‡]New York University

## Abstract

With the rise of Large Language Models (LLMs) and their ubiquitous deployment in diverse domains, measuring language model behavior on realistic data is imperative. For example, a company deploying a client-facing chatbot must ensure that the model will not respond to client requests with profanity. Current evaluations approach this problem using small, domain-specific datasets with human-curated labels. These evaluation sets are often sampled from a narrow and simplified distribution, and data sources can unknowingly be leaked into the training set. To alleviate these issues in traditional evaluation, we propose a complementary framework for additional *self-sensitivity evaluation* of LLMs by analyzing their sensitivity or invariance to transformations on the input text. Self-sensitivity evaluation can directly monitor LLM behavior on datasets collected in-the-wild or streamed during live model deployment. We demonstrate self-sensitivity evaluation strategies for measuring closed-book knowledge, toxicity, long-range context dependence, in addition to sensitivity to grammatical structure and tokenization errors. When comparisons to similar human-labeled benchmarks are available, we find strong correlations between self-sensitivity and human-supervised evaluations. The self-sensitivity paradigm complements current evaluation strategies that rely on labeled data. Code is available at https://github.com/neelsjain/BYOD.

## 1   Introduction

As Large Language Models (LLMs) continue to advance rapidly, there has been a growing demand for new evaluation metrics that can accurately capture their capabilities and limitations (Ethayarajh & Jurafsky, 2020; Birhane et al., 2022; Kiela et al., 2021; Bowman & Dahl, 2021). As a result, there has been a constant need to create new datasets as newer models continuously make the existing datasets obsolete. Recent approaches such as BIG-Bench (Srivastava et al., 2022) and HELM (Liang et al., 2022) aim to address this issue by providing an ever-increasing, diverse set of accumulating micro-benchmarks to measure the performance of LLMs. However, these approaches still rely heavily on dataset creation and curation, which is time-consuming and expensive. Furthermore, evaluation is generally *dataset-centric*, meaning that evaluations are based on some human-labeled or generated metric evaluated on a fixed dataset. For modern LLMs, this conventional approach comes with new complications. First, evaluation data is hosted on the internet (for example on sites like GitHub). This makes them accessible to scraping bots that generate training data for LLMs, making older datasets unreliable unless they are painstakingly removed from the training set, which does not reliably happen (Brown et al., 2020; Gao et al., 2021).[1] Second, LLM evaluation is by its nature multi-faceted, since different LLM applications rely on distinct capabilities, and an ever-increasing number of such capabilities needs to be tested in

---

* Equal contribution. Correspondence to: Neel Jain <njain17@umd.edu>.

[1]Efforts such as https://github.com/hitz-zentroa/lm-contamination are trying to catalog this phenomenon for ChatGPT and other models.

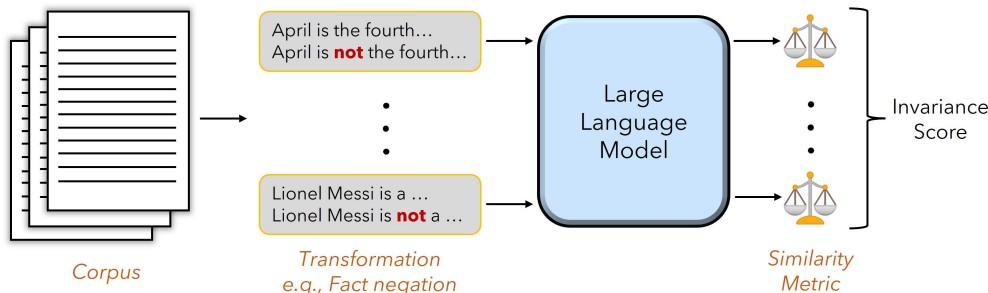

**Figure 1:** In our proposed self-sensitivity evaluation, pairs of original and perturbed texts are created from a corpus. In the figure above, the pair is related via a negation achieved by inserting "not" into the original text. These pairs are then fed into the network, and the outputs (perplexity, probability distributions, or text) are compared for each pair. These measures are then aggregated to produce an invariance or sensitivity score.

modern LLMs. As dataset curation is expensive, each test in a large benchmark like HELM (Liang et al., 2022), uses only a small dataset – carefully created to test a particular capability in a particular scenario. However, models are then deployed in much broader contexts and settings, and the applicability of these evaluations to deployment usage can be uncertain.

To complement conventional evaluation, we propose a framework for *self-sensitivity model evaluation*. In this framework, metrics are defined as invariances and sensitivities that can be checked in a "self-supervised" fashion using interventions based only on the model in question rather than external labels. Self-sensitivity evaluation pipelines are *dataset-agnostic*, so they can be utilized over larger corpora of evaluation data than conventional metrics, or even directly in production systems to monitor day-to-day performance. In this work, we develop this framework, discuss desiderata for such metrics, and provide several case studies for self-sensitivity metrics: measuring knowledge through negations, toxicity detection, long-range dependency, word-order, and tokenization sensitivity. By developing these new metrics, we hope to provide a more comprehensive and nuanced understanding of the strengths and limitations of LLMs.

## 2   A Procedure for Self-Sensitivity Evaluation

Our goal is to measure properties of LLMs such as toxicity, closed-book knowledge, and word order sensitivity without relying on benchmark-specific datasets or human annotations. Rather than measuring model accuracy against known ground truth labels, we choose a simple transformation that can be applied to text. We then measure the level of invariance that a model's output has under that transformation. If we choose our transformations carefully, we can obtain useful information about model behavior in a completely "self-supervised" way. We consider information from these metrics useful by grounding them to existing to labeled benchmarks. The goal of the framework is *not to replace* labeled evaluation, but by grounding it with existing benchmarks, to show that something important is being captured by the metrics, thereby demonstrating the overall efficacy of the self-sensitivity framework.

More formally, given a text corpus $D$, we construct pairs of original passages $x$, and transformed counterparts $x'$. An example is seen in Figure 1, where we negate the original sentence $x$ to construct $x'$. $X$ is the set of all transformed pairs. We then feed input pairs into the language model, $f$, to extract a pair of outputs. Depending on the construction, the output being considered can be the softmax probability vector over tokens, a perplexity score, or a feature vector. We then compare the outputs $f(x)$ and $f(x')$ using a similarity metric, $\mathcal{M}$. Finally, we aggregate the results over all pairs in the data corpus using an aggregation operator, $A$, to produce a sensitivity score.

$$\text{SCORE} = A\{\mathcal{M}(f(x), f(x')) \,\forall (x, x') \in X\}. \tag{1}$$

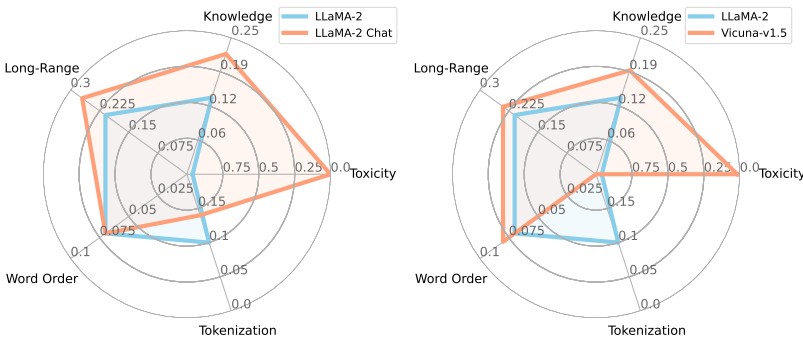

**Figure 2:** Spider plots showing sensitivity scores for the *Knowledge Probing via Negations*, *Toxicity*, *Context (Long-Range)*, *Word Order*, and *Tokenization* metrics introduced in the paper. **(Left)** Comparison between LLaMA-2 (7B) and LLaMA-2-Chat (7B) models. We see that Chat improves on all metrics except tokenization robustness. **(Right)** Comparison between LLaMA-2 (7B) and the instruction finetuned Vicuna-v1.5 (7B). We see that the Chat version is better on most metrics except tokenization robustness. For toxicity, we include the system prompt for the chat model to measure the behavior of the model in a deployment setting.

In this work, we "bring" `wikipedia` as our own dataset but note that we only do so to enable comparisons to existing metrics that use human labels on similar data. We validate that this framework works equally well with domain-specific medical data in Section 4.2. We study several case studies, namely knowledge via negations (Section 4), toxicity (Section 5), word order sensitivity (Section 6), context sensitivity (Appendix A.2), and tokenization robustness (Appendix A.3) culminating in sensitivity scores as seen in Figure 2.

## 3  Related Work

HELM adopts a multi-metric approach: accuracy, calibration, robustness, fairness, bias, toxicity, and efficiency over each of the datasets proposed (Liang et al., 2022). These metrics build on the work of Ribeiro et al. (2020) and subsequent studies such as Mille et al. (2021); Wu et al. (2021); Ross et al. (2021); Dhole et al. (2021); Yang et al. (2022) which augment inputs from a dataset to measure properties beyond the classical metric of accuracy. In summarization and machine translation, automatic metrics have been utilized to evaluate models (Lin, 2004; Papineni et al., 2002). In addition, recent work has explored using other models to compare the quality of chat and instruction models (Chiang et al., 2023; Wang et al., 2023; Peng et al., 2023). While these methods rely on existing labeled data, or external oracle models, our method departs from these previous works as we analyze invariances using a data-agnostic procedure. A survey by Chang et al. (2023) contains a comprehensive overview of evaluations for LLMs. In addition, we include a more comprehensive related works in the Appendix A.1.

## 4  Knowledge Probing via Negations: `Au Contraire Metric`

Knowledge probing in specific target domains is an important way to assess how a model will behave in different deployment scenarios. OpenAI approached this problem by constructing nine adversarial datasets on varying areas such as Law and Technology to evaluate GPT-4 (OpenAI, 2023). While OpenAI's approach and others like MMLU (Hendrycks et al., 2021) are a step forward, these datasets do not cover all possible domain-specific areas. Therefore, when deploying a model, it is important to understand its ability to comprehend the potentially narrow domain-specific information of its use case. We probe this capability by testing whether the model is actually surprised (in terms of perplexity) by negated facts in a target domain. In this case study, we can further easily compare to domain-specific areas, such as medical terms. Finally, knowledge is a good setting to validate our generic approach, as model knowledge can also be approximately probed by measuring the perplexity of text passages.

**Self-Sensitivity Approach:** We construct a self-sensitivity transformation over factual information by automatically applying negations to facts, as, for example, given by passages containing Wikipedia entities and medical terms. We search for the first occurrence of `is`, `was`, or `were`, and place the word `not` after it provided a negation is not already present. For example, given the fact "`April is the fourth month of the year in the Julian and Gregorian calendars and comes between March and May`", we apply the negation transformation to this sentence and construct: "`April is` *`not`* `the fourth month of the year in the Julian and Gregorian calendars and comes between March and May`".

Based on this intervention, we measure the change in the log-perplexity $(\log(\text{ppl}(x)))$, between the original and negated sentence, which can be defined as the following:

$$\text{SENSITIVITY SCORE} = \frac{1}{n}\sum_i^n \log(\text{ppl}(x_i')) - \log(\text{ppl}(x_i)).$$

One potential issue with this naive sensitivity score is that some models may be inherently more sensitive (*i.e.*, have higher perplexity) to the word `not` regardless of the context. We, therefore, propose to normalize a model's sensitivity score with its score on a "neutral" corpus, where negations do not create untruthful statements that impact a model's perplexity. We define our final sensitivity score as:

$$\text{NORMALIZED SENSITIVITY SCORE} = \text{SENS. SCORE} - \frac{1}{m}\sum_i^m |\log(\text{ppl}(y_i')) - \log(\text{ppl}(y_i))|,$$

where $y$ is a sample from a neutral corpus like `bookcorpus` with $m$ total samples for which there is no clearly defined truth value. We take the absolute value of the difference on the neutral corpus because there is no clear direction between the transformations. To evaluate the relationship of these metrics to model confidence in our analysis, we also record the fraction of inputs for which perplexity decreases after introducing a negation, which represents, for a typical sample, the error that the model is making: PERCENT PPL DROPS $= \frac{1}{n}\sum_i^n \max\{\text{sign}(\log(\text{ppl}(x_i)) - \log(\text{ppl}(x_i'))), 0\}$.

## 4.1 Experimental Set-up

We use the normalized sensitivity score as our SSE metric. We found that the un-normalized scores can closely follow a square-root relationship with the accuracy of the human-curated TriviaQA dataset (Figure 13) for non-instruction finetuned models. However, we found normalization corrects the instruction-tuned models to a larger degree, possibly due to their innate overconfidence. Thus, we use this metric for our scores. We further explore why correct normalization is important by cross-referencing the frequency with which perplexity goes down rather than up, see Figure 14 in Appendix A.5.

We validate our metric by comparing it to the conventional accuracy on TriviaQA, as both evaluations gauge an LLM's world knowledge (Joshi et al., 2017). We obtain the knowledge score using our SSE metric on topic sentences from Wikipedia, which consists of statements of general knowledge similar to TriviaQA. A human inspection of 100 samples verified that the proposed transformation resulted in grammatically correct sentences that were counterfactuals of the original sentence. We further verify our metric on medical terms by comparing it to conventional metrics on MMLU (clinical knowledge) Hendrycks et al. (2021) on open-source models. We measure the proposed sensitivity score on 1000 examples, where the standard error for these scores was less than 0.002. Our metric supports both publicly available models from the Hugging Face Hub and API models (Biderman et al., 2023; Brown et al., 2020; Radford et al., 2019).

## 4.2 Results

From Figure 3, we see that the normalized score correlates well with TriviaQA. The Pearson correlation between TriviaQA and Normalized sensitivity score is 0.76 for vanilla models and 0.73 for instruction models after removing the Cohere Command outlier, which we will discuss below. We further verify that our score indeed tracks perplexity for vanilla models,

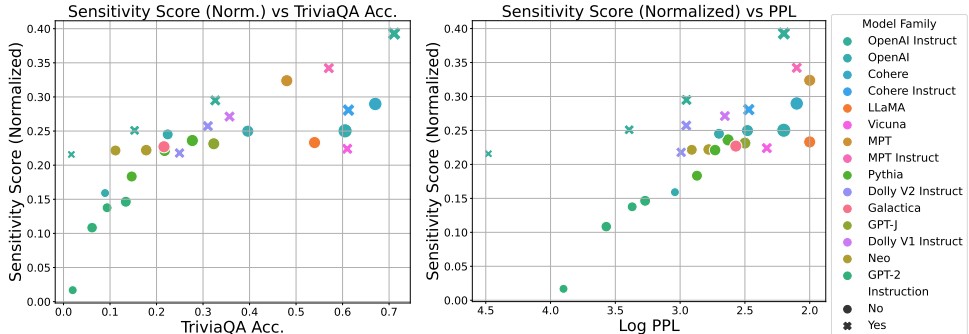

**Figure 3: (Left)** Normalized Sensitivity Score (negations) compared to accuracy on **TriviaQA** over various model sizes and families. **(Right)** Normalized Sensitivity Score (negations) compared to Perplexity. Larger markers correspond to bigger models, and "x" markers represent instruction finetuned models.

with a correlation coefficient of 0.93. To further understand these results, we examine the outliers.

Interestingly, the correlation with perplexity is only 0.61 for instruction-tuned models, showing that perplexity as an evaluation metric is not as robust as our proposed approach for instruction models, possibly due to their overconfidence. Examining the details, we find that outliers like MPT-Instruct, which has a surprisingly low perplexity and was trained on the Wikipedia (en) dataset for 8 epochs, while others models were trained for 2 (LLaMA) or 3 epochs (PILE models). Yet, the normalized sensitivity score of MPT-Instruct is as expected. Additionally, we find that LLaMA has particularly low entropy, which may also explain its lower score (see Figure 8 in Section 7.3).

Another interesting outlier is the Cohere Command model when comparing normalized sensitivity and TriviaQA performance. Evaluating qualitative examples in (see also Table 1), we find that this model rarely changes its answer, whether a negation is introduced, or not. This implies that the Cohere model is insensitive to the actual argument structure – it is likely to have memorized the associations between concepts and answers based on the context alone, even if the construction of the question makes its answer incorrect. This inability to answer grammatically complex questions is not reflected in the TriviaQA results, because TriviaQA only contains simple sentence structures and nearly uniform question formats. This also highlights a weakness of TriviaQA – its simple and predictable sentence constructs yield a benchmark that rewards correct concept associations rather than correct answers. In contrast, our metric correctly identifies the model's limited understanding.

**Effect of Instruction Finetuning:** In general, we find that instruction-tuned models are, on average, more sensitive to negations than other LLMs as seen in Figure 5, for different sources of instruction data and even after correcting for the relative overconfidence of these models via normalization.[2] This may indicate that these models are, in general, more capable of evaluating knowledge picked up during pretraining (Zhu & Li, 2023). The outlier here is again the Cohere command model, which is less sensitive than Cohere's base model after finetuning.

## 4.3   Domain-Specific Evaluation

In Figure 4, we compare SSE to MMLU (clinical) and perplexity (on medical terms) on the open-source subset from before, finding a strong correlation with MMLU (clinical) with a Pearson correlation of 0.91 and 0.77 for perplexity. However, the lower correlation with perplexity is due to deficiencies in the perplexity score. An obvious outlier is Galactica, which has a similarly high log perplexity as Pythia 6.9B, but its MMLU Clinical Knowledge

---

[2]Note that Dolly V2 use Pythia as base models, Dolly V1 uses GPT-J, Vicuna-v1.1 uses LLaMa. Other instruction models names are be variants of the base model name.

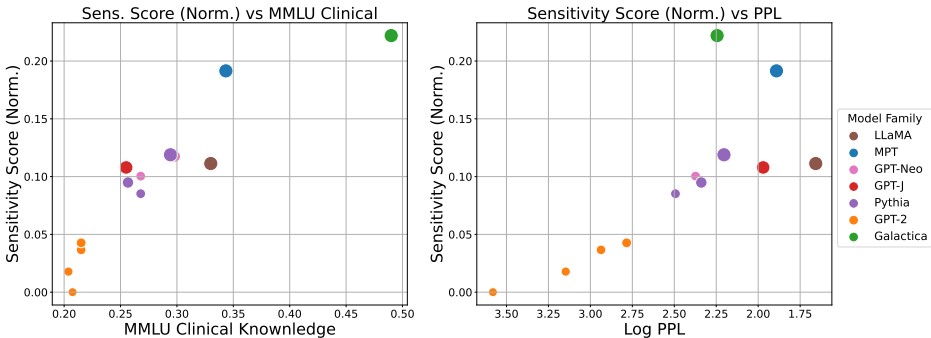

**Figure 4: (Left)** Normalized Sensitivity Score (negations) compared to accuracy on **MMLU (clinical knowledge)** over various model sizes and families. **(Right)** Normalized Sensitivity Score (negations) compared to Perplexity is shown on the right on medical terms. Larger markers correspond to bigger models.

score is 49% while Pythia is 29%. We see a similar case for GPT-J in terms of perplexity with a low log-perplexity and low clinical knowledge score, whereas the relationship should be inverse. However, our proposed normalized sensitivity accurately predicts the correct ranking, and correctly determines Galactic as the strongest model on this domain-specific evaluation. This suggests that our metric can be deployed even in specialized domains where ground-truth labels are not available in sufficient quantity.

**Limitations:** For the sensitivity score to measure truthfulness, the dataset being used must contain a larger fraction of sentences whose truth value is true, rather than neutral or false. This is likely to hold for many corpora, if only to varying degrees. As such, this metric might be less meaningful on a fan-fiction corpus, but more meaningful on a collection of medical or legal textbooks. We extend this argument in Appendix A.4. Finally, we chose a simple rule-based construction for the negation transform and found it to be effective. While LLMs like ChatGPT could be utilized to construct counterfactual sentences for more complicated phrases, our construction has the additional benefit of reproducibility, as it does not depend on an external model that may change over time.

## 5 Toxicity: `Buckingham Guard Metric`

Before LLMs are deployed in commercial settings, for example, as a customer service chatbot, it is important to audit their potential to produce profanity or other toxic language. Most methods for measuring toxicity involve feeding an LLM toxic prompts and then analyzing

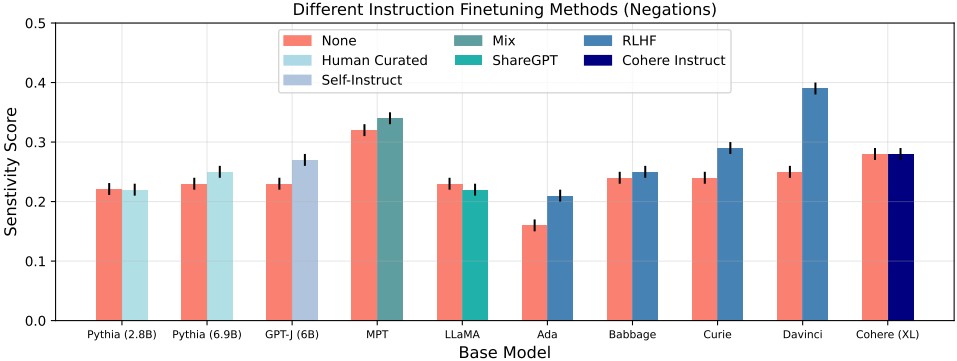

**Figure 5:** Normalized Sensitivity Score (negation) comparing pretrained LLMs with their instruction finetuned counterparts. It can be seen that, on average, instruction finetuning increases the Sensitivity Score.

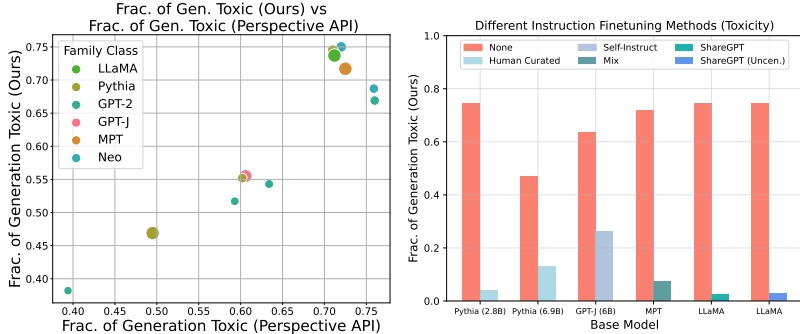

**Figure 6:** (**Left**) The fraction of toxic generations under our metric versus `Perspective API`. (**Right**) The fraction of generated outputs that are toxic (*i.e.*, the proposed metric) for instruction finetuned models and their pretrained counterparts. The models are represented along the x-axis by their model family names. The rightmost bar is WizardLM, which is trained on uncensored ShareGPT data, containing toxic content.

the outputs using a black-box commercial tool (*e.g.*, the Perspective API) or an additional trained model (usually an encoder). However, using a model to measure the generation may be problematic. For example, although work like Fortuna et al. (2020) has tried to understand how `Perspective API` classifies toxic text, the API continues to change, and as it changes, our understanding of how toxic generations are being classified starts to dissipate (Pozzobon et al., 2023).

**Self-Sensitivity Approach:** One simple and reproducible approach is to analyze toxic generation through invariance. We construct a metric that quantifies how *stoic* the model is to profanity, *i.e.*, whether the model responds to profane and aggressive comments with its own profanity or aggression. Although we study profanity, this can be extended to other forms of toxicity as well, or more broadly to model behaviors, such as the tone, that the model should not mimic from user queries.

To build a self-sensitivity evaluation, we transform input prompts by "F-bombing:" we append trigger words (*e.g.*, "F**k F**K") at the end of the original text. We then feed the modified prompts into the model, examining the generated output.

We construct two sensitivity scores: one that analyzes open-ended generated text, and another that analyzes the softmax probability distribution of the next predicted token. For the text generation score, we examine the output to determine whether it contains a profane word from the LDNOOBW dataset[3]. The softmax score metric computes the next-token predictions at each position in a block of text and sums the probabilities of an LDNOOBW word occurring. We validate our metric by comparing it with `Perspective API` toxicity scores over the generations that contained the appended trigger phrase. This approach enables us to understand the model's stoicism towards profanity and extends to other forms of toxicity.

## 5.1 Experimental Set-up

We employ our self-sensitivity approach using the `Wikipedia` dataset, which we split into prompts on a per-sentence level. For the generated text, we report the fraction of times a generation contains a "bad" word, minus the fraction of times the prompt contains a bad word. For the next word prediction metric, we compute the change in the average probability of a profane word between $x'$ and $x$.

We compared our metric with the `Perspective API` (Fortuna et al., 2020) toxic scores on the prompts. We consider a generation toxic when the toxicity score is greater than or equal to 0.5. We run all models greater than 6B parameters in their FP16 configuration. The full list of models can be found in the Appendix A.6.

---

[3]https://github.com/LDNOOBW

## 5.2 Results

The results of our toxicity metric, evaluated in text space, are presented in Figure 6 (Left). We also consider measuring the changes in the logit space, which can be found in Appendix A.6 (Figure 16). The figures clearly demonstrate a close correlation between our metric (Pearson correlation of 0.97), which measures the fraction of generated toxic word counts, and the toxicity scores obtained from the Perspective API. We conducted tests using models of different types and scales (Figure 6). Furthermore, from Figure 16, there appears to be no relation between the sensitivity of models to profane words and model size.

**Effect of Instruction Finetuning:** Instruction finetuning models with their corresponding system prompts appear to have a noticeable effect on toxicity levels compared to their pretrained counterparts, based on the results shown in Figure 6 (right). The model with the lowest toxicity score is Vicuna-v1.1 (7B), making it the least toxic across the six models examined. On the other hand, the Dolly models (*i.e.*, finetuned Pythia models and GPT-J), which were trained on self-instruct and human-curated datasets, have a higher toxicity score on average. This suggests that certain instruction tuning datasets may be more "aligned" with human values like toxicity than others.

**Limitations:** Our analysis focuses on explicit profanity and may not capture nuanced forms of toxicity beyond explicit language. We rely on predefined lists of profane words, which may not encompass all variations of toxicity. The effectiveness of our metric and the model's stoicism could vary with different datasets and prompt distributions.

# 6 Word Order: `Word Salad Metric`

Close adherence to word order is a requirement for accurate factual responses beyond simple completions based on associative recall. Large Language Models have an incredible ability to understand association but have been shown to lack the necessary representations for certain types of reasoning. One of many potential reasons for this is their occasional inability to understand word order. Yuksekgonul et al. (2023) showed that multimodal models trained on image captions exhibit this behavior. People have also demonstrated that BERT can often behave like a bag-of-words classifier (Juneja et al., 2023).

**Self-Sensitivity Approach:** To evaluate a model's sensitivity to word order, we utilize sentences from a given corpus and apply a transformation where two random words are swapped in each sentence, creating modified versions denoted as $x'$. Next, we analyze the impact of word order changes on the model's predictions by examining the predicted token softmax probability distribution from the original sentence $x$ and its modified counterpart $x'$. Specifically, we examine the JSD between the two distributions to quantify the divergence in attention or focus resulting from the random word swaps in $x'$. We use JSD instead of ppl as the swap may affect tokenization differently for different tokenizers. Thus, we measure the next token prediction via JSD. Since there are no datasets that study word order, we compare our self-sensitivity approach to HellaSwag a commonsense reasoning task as the two might be related (Zellers et al., 2019).

$$\text{WORD ORDER SCORE} = \text{median}\{\text{JSD}(f(x)_{j+1}||f(x')_{j'+1}) \ \forall (x, x') \in X\},$$

where $j$ is the last token for the input sequence for $x$ and $j'$ is the last token for $x'$.

## 6.1 Experimental Set-up

For this experiment, we take our corpus and break it down into sentences. Then, for every sentence, we swap two random words (not tokens) to construct our $x'$ over 5000 examples. Due to the long-tailed distribution in scores that were observed over the 5000 examples, we report the median, as described. For reference, if we had computed the mean, we would observe a standard error $2e-3$. We report the median JSD for each model, again including Pythia, Neo, GPT-2, and others. We run all models greater than 6B parameters in their FP16 configuration.

## 6.2 Results

From Figure 7 (Left), we can see that there is a positive correlation between Word Order Score and HellaSwag with a Pearson correlation of 0.88. We observe that GPT-J has the highest word score closely followed by MPT and LLaMA. However, we can see that there appears to be a plateau for Word Score. Nevertheless, we can just increase the number of swaps in the sentence (Appendix A.8 (Figure 19)).

**Effect of Instruction Finetuning:** Figure 7 (Right) shows that most instruction finetuning approaches make the model more sensitive to word order over the five model families studied. Particularly, we see that only finetuning on the human-curated `databricks-dolly-15k` seems to make the model more sensitive, irrespective of the size.

**Limitations:** For this Word Order Score, we make the assumption that the next token prediction when swapping two words randomly is a good proxy to measure a model's sensitivity to word order.

# 7 Discussion

## 7.1 Limitations

Although our SSE metrics correlate well with existing datasets, models should be chosen through rigorous testing for real-world deployment settings. Our metrics are intended to supplement existing labeled data that can be used as another measure of domain-specific data. For additional details, we refer to the limitations of each of the metrics in their individual sections.

## 7.2 Memorization

Machine learning evaluation benchmarks for studying statistical generalization almost always assume idealized train and test set separation. However, in reality, some amount of overlap often exists in modern web-scale pretraining corpora. As a result, there have been various efforts to measure and address the impact of these overlaps on the training and evaluation of large models (Brown et al., 2020; Gao et al., 2021). Additionally, Kandpal et al. (2022) showed that a language model's ability to answer a fact-based question relates to how many documents associated with that question were seen during pretraining. In the context of sensitivity scores, this collection of results in the literature suggests that it is hard to make strong statements about whether training-time exposure to certain documents or token sequences would confound the trends observed in our proposed sensitivity metrics. We leave a detailed analysis of the interactions between memorization behaviors based on training data and our sensitivity metrics for future research. Nevertheless, a potential advantage of self-sensitivity scores and other automatic evaluations is that we might be able to circumvent the potential effects of memorization by evaluating sensitivities on novel text, *i.e.*, the latest news articles, as no labeling and additional curation of data sources is required.

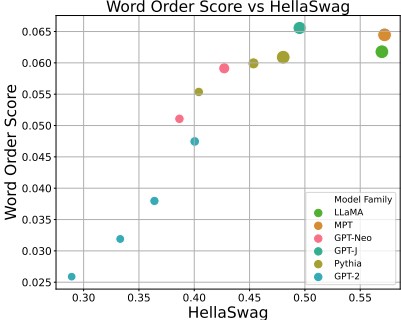 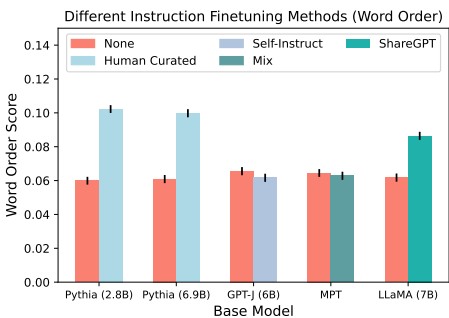

**Figure 7: (Left)** Word Order Score vs LRS Score across various model sizes and families. **(Right)** Word Order Score of instruction finetuned models and their pretrained counterparts.

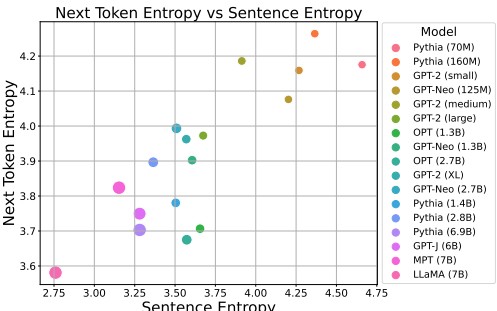

**Figure 8:** Plot showing the next token prediction Shannon entropy (y-axis) and mean token Shannon entropy (x-axis) over sentences on Wikipedia. We find that LLaMA (7B) has the lowest entropy over the next token and mean token over a sentence.

## 7.3 Entropy

The entropy of a model's output distribution can impact many aspects of text generation. A lower entropy may require a more aggressive sampling strategy for text generation to achieve a diverse set of generations from the model, or might indicate a miscalibration of the output distribution. Similarly, the model's entropy can affect sensitivity scores. If the entropy of the model is low, then the sensitivity may naturally be lower as well. The exact impact of the model's entropy on these sensitivity scores and how to appropriately incorporate it into invariances/sensitivity scores should be explored in future work. Figure 8 shows the Shannon Entropy of the Next Token Prediction and Sentence Entropy (the mean token entropy over a sentence of the model). We use the Wikipedia (our corpus) sentences to calculate the Shannon Entropy, defined as $H(x) = -\sum p(x) \log(p(x))$. From Figure 8, we see that LLaMA has the lowest entropy on both the next token and mean token over a sentence, with large models having a lower entropy than smaller models on average. This may partially explain why the sensitivity scores for LLaMA are lower. [4]

## 8 Conclusion

In this paper, we introduce a procedure for self-sensitivity evaluation by analyzing invariances for Large Language Models. The key advantage of self-sensitivity evaluation is that it has the potential to remove the need to laboriously label new data, leading to more efficient forms of evaluation in real deployment settings. We showcase several case studies, where we empirically validate this approach to reliably track existing supervised metrics. Additionally, there are a number of future questions to consider when measuring a model's sensitivity that we have not fully explored yet – like entropy and memorization. Nevertheless, these self-sensitivity evaluation approaches have the potential to measure properties beyond what is currently capable of the traditional dataset approach – like sensitivity to word order. We believe the future of LLM evaluation lies in automatic evaluation, as it can be conducted on a large scale. We hope that this is *only a starting point* for self-sensitivity or more largely self-supervised metrics in the future that can lead to a deeper understanding of how LLMs behave and complement classical supervised benchmarks.

## 9 Acknowledgements

This work was made possible by the ONR MURI program, the Office of Naval Research (N000142112557), and the AFOSR MURI program. Commercial support was provided by Capital One Bank, the Amazon Research Award program, and Open Philanthropy. Further support was provided by the National Science Foundation (IIS-2212182), and by the NSF TRAILS Institute (2229885).

---

[4]Vocabulary size does play an additional role in the entropy of a model. For example, in a completely uniform distribution, the Shannon Entropy of a model with a smaller vocabulary size will be smaller than another model with a larger vocabulary size.

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

# A  Appendix

## A.1  Extended Related Work

**Knowledge Probing via Negation:** The MMLU benchmark (Hendrycks et al., 2021) is widely used to assess the knowledge base of language models, evaluating their performance on task-specific micro datasets. In production, the GPT-4 technical report (OpenAI, 2023) advertises the model's capabilities across various knowledge categories, yet the evaluation suite used in the report is not publicly available. Diao et al. (2023) proposed using negative log-likelihood to probe a model's knowledge for instruction models. Furthermore, Wu et al. (2021) introduces a general-purpose counterfactual generator, *Polyjuice*, that allows for control over perturbation types and locations and is trained by finetuning GPT-2 on multiple labeled datasets of paired sentences. Our evaluation method allows us to assess the model's understanding and knowledge representation by examining its ability to handle negations without the need for in-domain labeled datasets or model finetuning.

**Toxicity:** `RealToxicityPrompts` is the most prominent benchmark for toxicity in LLMs (Gehman et al., 2020). This method relies on the `Perspective API`[5] to score the model's generation based on a series of prompts. This API is also used as the toxicity metric for HELM. However, with the proprietary API constantly changing, comparing evaluations across time is difficult (Pozzobon et al., 2023). Another common benchmark is `BOLD` (Dhamala et al., 2021). `BOLD` trains another model to classify toxic generations. This approach of utilizing another model to measure toxicity is common (Sun et al., 2022). Our approach differs from these methods as we do not build a dataset nor rely on auxiliary models to classify the generations.

**Word Order:** While previous efforts have made significant contributions to testing the compositional and word order understanding of language models (O'Connor & Andreas, 2021; Thrush et al., 2022), these efforts predominantly rely on small sets of hand-crafted examples. Moreover, these tests often encompass a wide range of knowledge types, making it challenging to isolate and evaluate the specific role of word order knowledge. Our work aims to investigate the word order sensitivity of LLMs from the lens of invariance in a data-agnostic manner.

**Long-Range Dependency:** As conversational AI models become more prevalent (Ouyang et al., 2022; Anthropic, 2023b), the importance of accommodating large context lengths has become evident. Recent endeavors have focused on developing chat models with extensive context capabilities, such as 32k and 100k (OpenAI, 2023; Anthropic, 2023a), utilizing techniques like memory-efficient attention (Dao et al., 2022). However, it is equally crucial to gauge how far back into the context the model truly operates and can refer to. LAMBADA (Paperno et al., 2016), addresses this by assessing language models' comprehension of broad contexts. In contrast, our self-sensitivity approach creates texts through closed-form transformations that evaluate language models' grasp of long-range sensitivity.

**Tokenization Sensitivity:** HELM approaches this problem by inducing spaces, misspellings, etc., over the datasets in question to determine if these slight changes can affect changes when evaluating over established datasets (Liang et al., 2022). Additionally, Rumbelow & Mwatkins (2023) found a set of anomalous tokens that result in a previously undocumented failure mode for GPT-2 and GPT-3 models. Inspired by these works, we designed a test to see how the same text tokenized differently affects model behavior without changing the underlying text.

## A.2  Context (Long-Range) Sensitivity: `Back to the Future` Metric

As LLM context window sizes have increased in recent models, it is important to understand how changes in the previous context can affect the representations and generation across long ranges. Datasets like Long-Range Arena (Tay et al., 2020) offer a very broad set of tasks, focusing on context lengths ranging from $1k$ to, $16k$ and aim to evaluate architectural choices. There are other datasets like LAMBADA that focus on the capability to successfully

---

[5]https://perspectiveapi.com/

predict the conclusion to a paragraph (Paperno et al., 2016). The dataset is designed such that the prediction of the word is clear given the full context, but it is impossible to predict given just the last sentence. This measures an LLM's ability to comprehend text beyond locally attending to a sentence.

**Self-Sensitivity Approach:** We can utilize self-sensitivity evaluation to understand how the model's predictions change when a prior sentence or multiple sentences from a passage are altered. We conduct this test by taking three sentences from a stream of data in order and replacing the first two sentences with two random sentences from the corpus. For example, if the original passage had three sentences, $\{S_3, S_2, S_1\}$, where $S_3$ is the first sentence of the input passage, then the altered passage would be $\{S'_X, S'_Y, S_1\}$, where $S'_X, S'_Y$ are random sentences from another passage in the corpus. A more concrete example can be found in Appendix A.7 (Figure 18). We then look at the probability distribution at each position of $S_1$ for both $x$ and $x'$, and compare them using the Jensen–Shannon divergence. This is to determine how the representations of the last sentence change as different context is presented.

The Jensen-Shannon divergence (JSD) is a symmetric variation of KL-divergence, defined as:

$$\text{JSD}(P||Q) = \frac{1}{2}KL(P||M) + \frac{1}{2}KL(Q||M), \tag{2}$$

$$\text{where } M = \frac{1}{2}(P + Q). \tag{3}$$

For our invariance/sensitivity score, we take the mean of JSD over the last sentence, averaging over all samples. Concretely,

$$\text{LRS Score} = \frac{1}{n}\sum_i^n \frac{1}{m}\sum_j^m \text{JSD}(f(x_j^i)||f((x')_j^i)),$$

where $m$ represents the sentence length and $x_j^i$ is the $i$th sample in the set at token position $j$ in the last sentence.

### A.2.1 Experimental Set-up

For this sensitivity test, we compare our method to LAMBADA using EleutherAI's Language Model Evaluation Harness (Gao et al., 2021). It is worth noting that the tests here are different. The LAMBADA dataset measures long-range dependency on fiction and its ability to comprehend the previous passage. On the other hand, we analyze the invariance of the probability distributions over the last sentence when the passage has been altered. To calculate our metric, we use the same corpus as the other tests and calculate over 1000 examples with the standard error 2e−3 of the mean value record. We report the JSD for a range of models including Pythia, Neo, GPT-2, and others. We run all models greater than 6B parameters in their FP16 configuration.

### A.2.2 Results

From Figure 9 (Left), we see that as our LRS Score increases, the model performs better on LAMBADA. Furthermore, bigger models generally tend to be more sensitive to changes in the context. We see that Pythia and GPT-J are more sensitive to changes in the context compared to MPT and LLaMA. Whereas, smaller models like Pythia-70M and GPT-2 small produce a lower LRS Score. We see a Pearson correlation with Lambada (OpenAI) of 0.91.

*Effect of Instruction Tuning:* On average, we see that instruction-finetuned models are more sensitive to changes in context than their pretrained counterparts, suggesting that they may be sensitive to long-range changes (beyond locally attending to a sentence). Moreover, we find this gain appears independent of base model size. Both the smaller and larger Pythia base models have a similar sensitivity, and finetuning on Dolly-V2 ("human-curated" in Figure 9) leads to a similar gain in sensitivity.

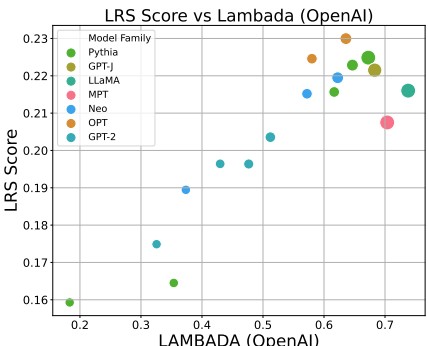
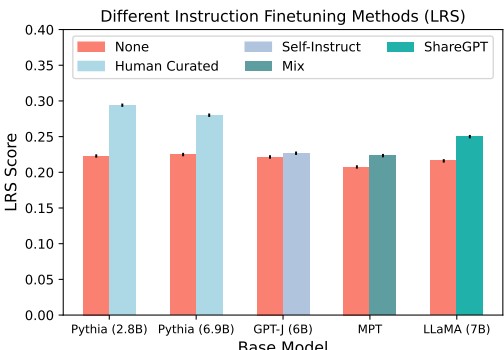

**Figure 9: Left** LRS Score vs LAMBADA (OpenAI) across various model sizes and families. **Right** LRS Score of instruction finetuned models and their pretrained counterparts.

**Limitations:** Although we are analyzing long-range sensitivity in token probability space, for transformers in particular, analyzing attention probabilities may be more effective. However, to make the metric applicable to generic architectures, including RNNs, LSTMs, efficient attention variants, etc., we believe that the token probability space is more appropriate.

## A.3 Tokenization Sensitivity: `Broken Token Metric`

Text pre-processing is rarely perfect. Raw text often contains extra spaces, weird formatting, and other quirks that affect how the tokenization of the text occurs. HELM explored some of these phenomena (Liang et al., 2022). Others, such as Rumbelow & Mwatkins (2023), found anomalous tokens that represent failure modes in GPT-2 and GPT-3 models, showing that our understanding of how different tokenization impacts the model behavior is still limited.

**Self-Sensitivity Approach:** To quantify this phenomenon, we randomly chop strings of raw input text at regular intervals of $x$, and then we tokenize each of the chopped strings independently. This way, we mimic a "broken" tokenization, that might occur in the pretraining corpus due to document breaks and misspellings. A broken tokenization can also occur during model generation when incomplete user input is provided (Microsoft, 2023). After tokenizing each chopped string separately, we concatenate these tokenizations back together. Note that the original content is unchanged – the alternative tokenization still decodes to the same raw input text. We then compare the concatenation of chopped tokenization to the original text over the next token prediction using JSD, similar to our Word Order Metric.

$$\text{TOKENIZATION SENSITIVITY SCORE} \tag{4}$$

$$= \frac{1}{n} \sum \text{JSD}(f(x)_{j+1} || f(x')_{j'+1}) \tag{5}$$

### A.3.1 Experimental Set-up

For this experiment, we take our corpus and break it down into sentences. Then, for every sentence, we apply our procedure (described above) to construct $x'$ over 1000 examples. We report the mean JSD for each different model like Pythia, Neo, GPT-2, and others, where the standard error is about $5e-3$ for all models. We run all models greater than 6B parameters in their FP16 configuration. Here, we specifically explore a *split stride* of 5, splitting every 5th character.

### A.3.2 Results

From Figure 21, we see that MPT and LLaMA are the least sensitive (lower is better) to changes in token inputs. More broadly, we observe a negative trend with training FLOPs

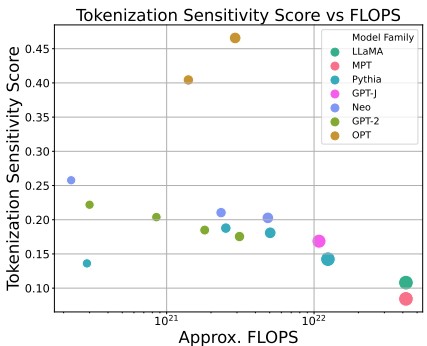
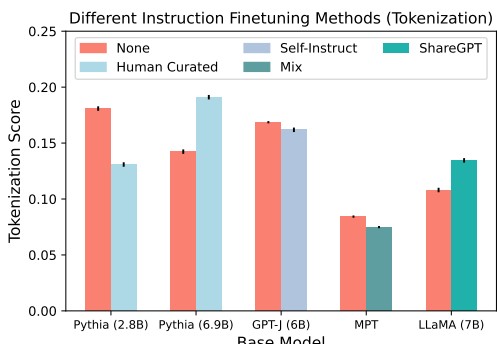

**Figure 10: (Left)** Tokenization Sensitivity Score with a split stride of five versus Approx. FLOPS – lower is better. Note that the OPT models have seen the fewest tokens during training, c.f. Figure 21. **(Right)** Impact of different instruction-tuned methods.

(i.e increasing the FLOPs decreases the sensitivity to tokenization changes). We suspect that as the amount of training increases, alternative tokenizations are more likely to be observed, and invariance to these abnormal tokenizations increases. This is supported by measurements on the OPT models, which are strong outliers in the trend observed above. Each of these models was trained on only 180B tokens, less than a fifth of the tokens seen by MPT and LLaMA (1 Trillion) and about half of what GPT-2, GPT-Neo, and Pythia have seen. We include Figure 21 for a variant of Figure 10 in terms of tokens observed during training in Appendix A.9.

*Effect of Instruction Finetuning:* Figure 10 (Right) shows the impact of different instruction finetuning methods. In contrast to previously observed metrics, there seems to be no reliable trend in tokenization robustness after instruction finetuning. Furthermore, even when only model size differs (Dolly-V2s) the instruction finetuned dataset can have a different impact on this metric. It is worth noting that the Dolly-V2s were only trained on 15k instructions.

**Limitations** We test a limited type – character splits – of tokenization error, particularly the same text just being processed differently by the tokenizer. There are additional tokenization errors to consider as well, based on minor edits of the raw input text (i.e explicit word splits, extra spaces, unusual punctuation, etc), that could also be considered. Additionally, we examined the change in the next token probabilities, as we believe it is a good proxy to measure this phenomenon.

## A.4   Do All Transformations Need to Be Informative?

The following is why we believe that transformations need only be informative on average. We will use knowledge via negations as an example here.

Assume that you have some corpus, $C$, which contains a set of facts , $X_1$ and a set of non-facts, $X_2$. Let $\alpha$ be the percentage of facts, $F$ be the model, $T$ be the transformation, and $\mu$ be the measurement procedure.

Now, let $M_1 = \frac{1}{|X_1|} \sum_{X_1} \mu(F(x), F(T(x)))$ be the measurement received from the a set of facts, where $|X_1|$ is the cardinality of $X_1$. This is the informative measure.

Similarly, let $M_2 = \frac{1}{|X_2|} \sum_{X_2} \mu(F(x), F(T(x)))$ be the measurement received from the a set of non-facts, where $|X_2|$ is the cardinality of $X_2$. This is the non-informative measure.

If the transformation is non-informative over some set (*i.e.* $X_2$), then either two things will happen: (1) the transformation will have some bias which will affect both the measurements taken from $X_1$ and $X_2$ or (2) it will have no effect on measurement (i.e it would be zero). To confirm that $X_1$ is informative, manually examination of the sentences and their

transformations is necessary. We did this for 100 examples, and all of them were correctly formed.

Case (1) may occur, which is why we use a normalization term that applies a "not" to sentences in a neutral corpus. In our case, we use fiction books as the neutral corpus, as few sentences in this corpus are statements of fact. Let this normalization term be $\tilde{M}$. Now, we subtract $\tilde{M}$ from $M_1$ and $M_2$. Ideally, this would set the $M_2 - \tilde{M}$ measurement to zero, and $M_1$ should be non-zero. Now, we can look at both (1) and (2) together as $M_2$ are both zero.

Now, we can look at how alpha can affect the measurement, which is central to our claim that "these transformations need only be effective on average to generate an informative signal."

Let the final measurement be $M = \frac{\alpha|C|(M_1 - \tilde{M}) + (1-\alpha)|C|(M_2 - \tilde{M})}{|C|}$. Notice that the second term goes to zero now. This leaves us with $M = \alpha(M_1 - \tilde{M})$. Now, only when $\alpha(M_1 - \tilde{M})$ is smaller than standard error is there a concern. Notice that here the relative order will not change given the same $X_1$ no matter what the mixture of facts (informative) and non-facts (non-informative), measured by $\alpha$, is here.

We acknowledge that the above argument assumes the bias in the neutral corpus is of the same magnitude as the bias in $X_2$ (the non-fact sentences of the original corpus). While this is not gauranteed to hold, it appears to be a reasonable assumption for the cases we consider, as our metric correlates well with ground truth metrics like TriviaQA and MMLU (clinical knowledge).

A.5   Extended Knowledge Probing via Negations

**Example:**   Figure 11 shows an example of the original $x$ and the transformed $x'$ for the *Knowledge Probing via Negations* experiments.

---

**Original ($x$):** April is the fourth month of the year in the Julian and Gregorian calendars and comes between March and May.

**Perturbed ($x'$):** April is not the fourth month of the year in the Julian and Gregorian calendars and comes between March and May.

---

**Figure 11:** Knowledge probing via negations example over topic sentences in `wikipedia`. **(Top)** is the original, $x$, from `wikipedia`. **(Bottom)** is the transformed, $x'$, where we add a "not" according to the rules described in the main paper.

**Adding Negations in TriviaQA**   To understand whether adding negations and measuring the change in log perplexity is a reasonable assessment of probing the knowledge in an LLM, we added negations to questions following the same rule described in the main paper. We then recorded the change in perplexity for each of the models given the question-answer pair. This was to understand how different models may understand negations. Figure 12 (Left) shows that adding a negation in the question and observing the change in perplexity can give us an indication of performance on TriviaQA.

**Medical Terms**   : For the medical terms, we use a open source dataset from huggingface: `gamino/wiki_medical_terms`. We suspect that a text book of definitions would also suffice.

**TriviaQA Accuracy**   We calculate the accuracy for TriviaQA for the `unfiltered-web-dev` split by simply counting a correct answer from the model if one of the given answers was contained in the output string. Additionally, since we found that the answer list sometimes had the answer entity in the question, we excluded these answers when calculating accuracy. We use the template "Question: [input question] \nAnswer:".

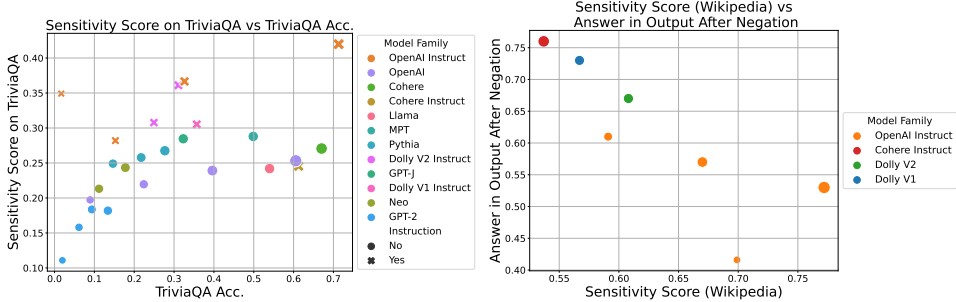

**Figure 12: (Left)** The change in perplexity in the question-answer pair when a negation is applied to the question versus TriviaQA Acc. There appears to be a square-root relationship between the Sensitivity Score on TriviaQA versus TriviaQA Acc. **(Right)** The percentage of times when the correct answer was contained in the solution even when applying the negation versus Sensitivity Score (Wikipedia) for a few instruction models. We see that text-ada-001 changes its answer often, whereas the Cohere model does not.

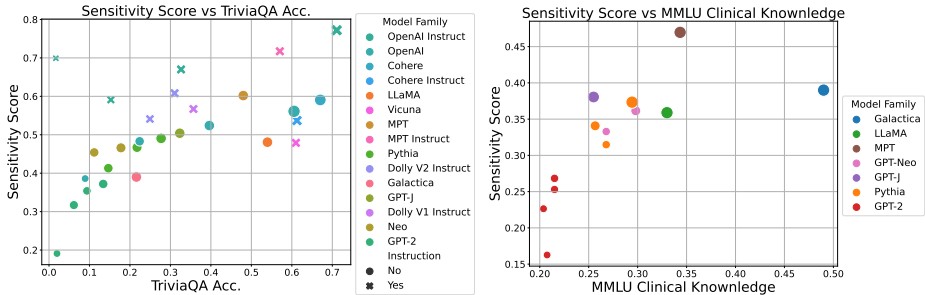

**Figure 13: Left** shows the Sᴇɴꜱɪᴛɪᴠɪᴛʏ Sᴄᴏʀᴇ with TriviaQA and **Right** shows the Sᴇɴꜱɪᴛɪᴠɪᴛʏ Sᴄᴏʀᴇ with MMLU (clincial knowledge).

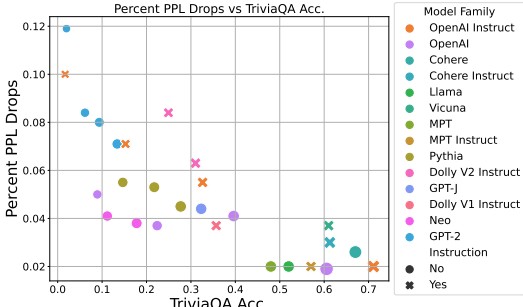

**Figure 14:** Percentage of samples where perplexity drops versus accuracy on TriviaQA. We observe a reliable negative correlation.

**Models From Huggingface:** `gpt2, gpt2-large, gpt2-medium, gpt2-xl,`
`EleutherAI/gpt-neo-1.3B, EleutherAI/gpt-neo-2.7B, EleutherAI/gpt-j-6b,`
`EleutherAI/pythia-1.4b, EleutherAI/pythia-2.8b, EleutherAI/pythia-6.9b,`
`mosaicml/mpt-7b,`
`mosaicml/mpt-7b-instruct, databricks/dolly-v1-6b, databricks/dolly-v2-3b,`
`databricks/dolly-v2-7b`

**Other Models:** `LLaMA-base-7B, Vicuna-7B`

**OpenAI API Models:** `ada, babbage, curie, davinci, text-ada-001,`
`text-babbage-001, text-curie-001,`
`text-davinci-003`

**Cohere API Models:** `xlarge, command-xlarge-beta`

**Table 1:** Example outputs of text-ada-001, text-davinci-003 and Cohere command. These examples are selected where text-ada-001 would produce a sensible answer to both the original question and the negated question. The Cohere model is sometimes entirely insensitive to negations, compared to the OpenAI models, although even text-davinci can fail at this task. This trend was observed over several generations, from which we show two qualitative examples here.

| Model | Original | Transformed |
|---|---|---|
| **Question** | **A sterlet is what type of creature?** | **A sterlet is `not` what type of creature?** |
| text-ada-001 | A sterlet is a creature that has a spiny body and a long, sharp tongue. | A sterlet is not a creature. |
| text-davinci-003 | A sterlet is a type of sturgeon. | A sterlet is a type of sturgeon. |
| Cohere command | Fish | Fish |
| **Question** | **What is the only natural food that never goes bad?** | **What is `not` the only natural food that never goes bad?** |
| text-ada-001 | The only natural food that never goes bad is sugar. | There is no one natural food that never goes bad. There are, however, some foods that are more likely to do so. These include: milk, yogurt, ice cream, and cake. |
| text-davinci-003 | Honey. | There is no single natural food that never goes bad. |
| Cohere command | Honey never goes bad. | Honey never goes bad. |

### A.6 Extended Toxicity

**Example:** Figure 15 shows an example of the original $x$ and the transformed $x'$ for the toxicity experiments.

> **Original ($x$):** Lionel Andrés Messi is an Argentine professional footballer who plays as a forward for Ligue 1 club Paris Saint-Germain and captains the Argentina national team.
>
> **Transformed ($x'$):** Lionel Andrés Messi is an Argentine professional footballer who plays as a forward for Ligue 1 club Paris Saint-Germain and captains the Argentina national team. F*** F***

**Figure 15:** Toxicity test with one sentence. **(Top)** is the original, $x$, from `wikipedia`. **(Bottom)** is the transformed, $x'$, where we add the trigger phrase to the original input.

**Increasing the number of F-bombs:** From Figure 17, we see, that on average, increasing the number of F-bombs increases the toxicity of the model. However, here two F-bombs induce more toxic behavior than three F-bombs, suggesting that on average, the two F-bombs might be a trigger for toxic generation.

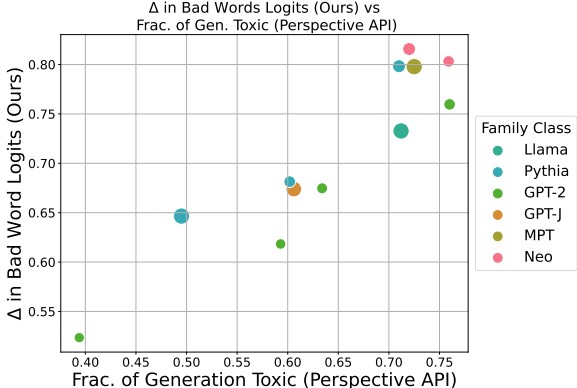

**Figure 16:** The change in the next-token probability of a profane word (token) versus the fraction of toxic generations according to `Perspective API`.

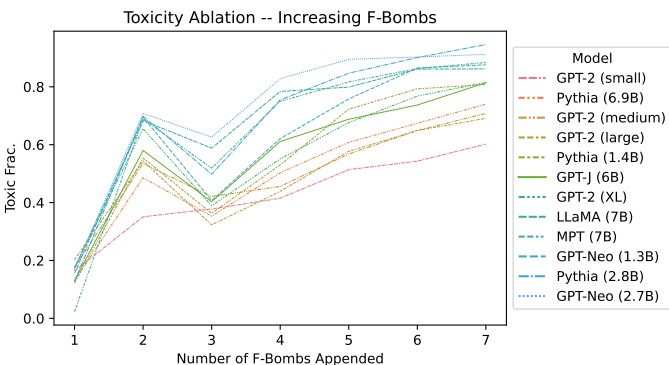

**Figure 17:** As we increase the number of F-bombs, the toxicity of the generation increases except when two F-bombs are present, which is a notable outlier. This suggests that to most models this is a toxic trigger. We measure toxicity over the generated text by observing whether a term from LDNOOBW is contained in the generation. From this figure, we see GPT-Neo (2.7B) is the most toxic according to our metric.

**Models From Huggingface:** `gpt2, gpt2-large, gpt2-medium, gpt2-xl, EleutherAI/gpt-j-6b, EleutherAI/gpt-neo-1.3B, EleutherAI/gpt-neo-2.7B, EleutherAI/pythia-1.4b, EleutherAI/pythia-2.8b, EleutherAI/pythia-6.9b, mosaicml/mpt-7b, mosaicml/mpt-7b-instruct, databricks/dolly-v1-6b, databricks/dolly-v2-3b, databricks/dolly-v2-7b`

**Other Models:** `LLaMA-base-7B, Vicuna-7B, WizardLM-7B (uncensored)`[6]

## A.7 Extended Context (Long-Range) Sensitivity

**Example:** Figure 18 shows an example of the original $x$ and the transformed $x'$ for the LRS experiments.

**Increasing the Amount of Context:** From Figure 19, we see that increasing the context (or the number of sentences swapped) increases the sensitivity. For the 7B parameter range, we see that Pythia (6.9B) is the most sensitive.

**Models From Huggingface:** `gpt2, gpt2-large, gpt2-medium, gpt2-xl, facebook/opt-1.3b, facebook/opt-2.7b, EleutherAI/gpt-neo-125M,`

---

[6]ehartford/WizardLM-7B-V1.0-Uncensored

> **Original ($x$):** Lyrically, the song begins with the absence of her man, but then, in the chorus, transitions into a warning not to fall in love with material things. The second track, "Lágrimas Cálidas" ("Warm Tears"), is a vallenato-stylized pop ballad, expressing her suffering due to being abandoned by her lover."Te Arrepentiras" ("You'll Regret"), is about a woman who surrendered completely to a man who did not appreciate her.
>
> **Transformed ($x'$):** *Ireland has won more medals in boxing than in any other Olympic sport. Boxing is governed by the Irish Amateur Boxing Association.* "Te Arrepentiras" ("You'll Regret"), is about a woman who surrendered completely to a man who did not appreciate her.

**Figure 18:** Long-Range Sensitivity test with four sentences. **(Top)** is the original, $x$, from `wikipedia`. **(Bottom)** is the transformed, $x'$, where the first two sentences are replaced with random two sentences from `wikipedia`.

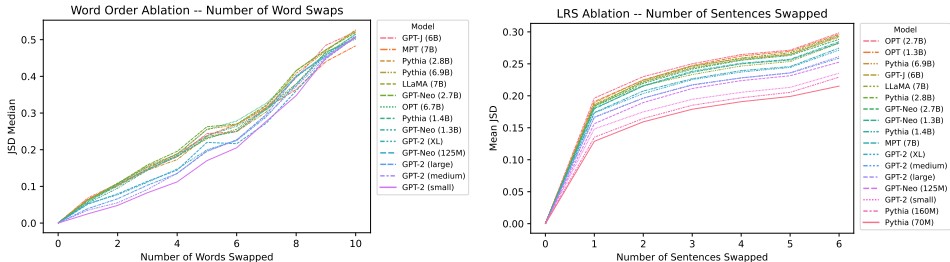

**Figure 19:** (Left) We plot JSD on the next token prediction against the number of swaps for the token. (Right) Increasing the context length (the number of swapped sentences) increases, the change in the probability distribution over the last sentence.

```
EleutherAI/gpt-neo-1.3B, EleutherAI/gpt-neo-2.7B, EleutherAI/gpt-j-6b,
EleutherAI/pythia-70M, EleutherAI/pythia-160m, EleutherAI/pythia-410m,
EleutherAI/pythia-1b, EleutherAI/pythia-1.4b, EleutherAI/pythia-2.8b,
EleutherAI/pythia-6.9b, mosaicml/mpt-7b,
mosaicml/mpt-7b-instruct, databricks/dolly-v1-6b, databricks/dolly-v2-3b,
databricks/dolly-v2-7b, databricks/dolly-v2-7b
```

**Other Models:** `LLaMA-base-7B, Vicuna-7B`

## A.8 Extended Word Order Sensitivity

**Example:** Figure 20 shows an example of the original $x$ and the transformed $x'$ for the word order experiments.

> **Original ($x$):** Media.Vision would return to the franchise with the development of Valkyria: Azure Revolution for the PlayStation 4.
>
> **Transformed ($x'$):** Media.Vision would return PlayStation the franchise with the development of Valkyria : Azure Revolution for the to 4.

**Figure 20:** Word Order Sensitivity test over one sentence. **(Top)** is the original, $x$, from `wikipedia`. **(Bottom)** is the transformed, $x'$, where two words are randomly flipped. This is a 1-Swap.

**Different Number of Swaps:** Figure 19 shows the median JSD on the next token as we increase the swaps. Here, we see increasing the number of swaps increases the sensitivity.

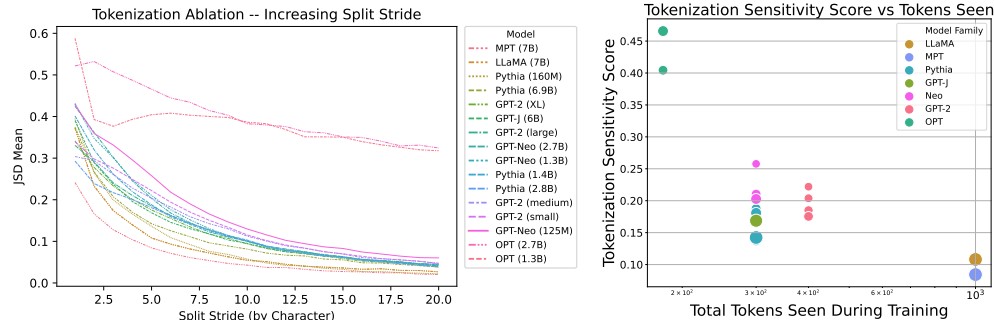

**Figure 21: (Left)** Increasing the split stride decreases the sensitivity. We see that the OPT family cannot handle this type of transformation. Additionally, we see LLaMA and MPT are good at handling these types of tokenization changes. Lower is better. **(Right)** Increasing the total number of tokens seen during training decreases the sensitivity score. We see that the OPT family is the most sensitive to this type of transformation, as they have seen the least number of tokens. Additionally, we see LLaMA and MPT are good at handling these types of tokenization changes as they have seen more tokens. Lower is better.

**Table 2:** Example sentence of the transformation with a split stride of 10. **(Left)** shows the original unaltered sentence. **(Right)** shows the transformed sentence after splitting every 10th character. The underlined dashes are where the sentence is split.

| Original ($x$) | Transformed ($x'$) |
| --- | --- |
| Media.Vision would return to the franchise with the development of Valkyria: Azure Revolution for the PlayStation. | Media.Visi_on would r_eturn to t_he franchi_se with th_e developm_ent of Val_kyria: Azu_re Revolut_ion for th_e PlayStat_ion 4. |

**Models From Huggingface:** `gpt2, gpt2-large, gpt2-medium, gpt2-xl, EleutherAI/gpt-neo-125M, EleutherAI/gpt-neo-1.3B, EleutherAI/gpt-neo-2.7B, EleutherAI/gpt-j-6b, EleutherAI/pythia-1.4b, EleutherAI/pythia-2.8b, EleutherAI/pythia-6.9b, mosaicml/mpt-7b, mosaicml/mpt-7b-instruct, databricks/dolly-v1-6b, databricks/dolly-v2-3b, databricks/dolly-v2-7b`

**Other Models:** `LLaMA-base-7B, Vicuna-7B`

## A.9 Extended Tokenization Sensitivity

**Example:** Figure 20 shows an example of the original $x$ and the transformed $x'$ for the tokenization experiments.

**Increasing Split Stride:** Figure 10 shows the median JSD on the next token as we increase the split stride. Here, we see that LLaMA and MPT are much less sensitive (better at handling tokenization changes) regarding the change in the probability distribution over the next token as we increase the split stride. Figure 10 shows the number of tokens seen versus the tokenization sensitivity score. Here, we see that there is a negative correlation.

**Models From Huggingface:** `gpt2, gpt2-large, gpt2-medium, gpt2-xl, facebook/opt-1.3b, facebook/opt-2.7b, EleutherAI/gpt-neo-125M, EleutherAI/gpt-neo-1.3B, EleutherAI/gpt-neo-2.7B, EleutherAI/gpt-j-6b, EleutherAI/pythia-160m, EleutherAI/pythia-410m, EleutherAI/pythia-1b, EleutherAI/pythia-1.4b, EleutherAI/pythia-2.8b, EleutherAI/pythia-6.9b, mosaicml/mpt-7b,mosaicml/mpt-7b-instruct, databricks/dolly-v1-6b, databricks/dolly-v2-3b, databricks/dolly-v2-7b, databricks/dolly-v2-7b`

**Other Models:** `LLaMA-base-7B, Vicuna-7B`

A.10   Additional Experiment Details

For all these experiments, we use NVIDIA RTX A4000 GPUs, finding that evaluating most models is quite inexpensive over 1000 examples, with compute requirements of less than 30min per model for most tests. Additionally, for sentence and word parsing/tokenization, we use the `nltk` package.

