# OpenReview forum: "Bring Your Own Data! Self-Sensitivity Evaluation for Large Language Models"
_colmweb.org/COLM/2024/Conference — COLM_

### Official Review · Reviewer_zqFn · 2024-05-04

**Rating:** 5
**Confidence:** 3
**Ethics Flag:** 1

**Summary:**

This paper introduces a straightforwardly designed SSE evaluation metric based on different perturbation methods to detect a model's self-sensitivity. Extensive validation reveals a high correlation between the assessment of self-sensitivity and direct evaluation of language models on benchmarks, thereby demonstrating the effectiveness of the self-sensitivity method for assessing unlabeled data.

**Questions To Authors:**

1. Figure 2 seems to missing the performance on the "Negations".
2. In section 4, the normalized sensitivity score is highly correlated with the "neutral corpus bookcorpus", which raise following questions
   1. Why is the bookcorpus used in this paper. Is wikipedia also useful?
   2. The proposed metric inevitably is influenced by multiple factors of data in the bookcorpus: (1) the number of **m** total samples; (2) the sampled data samples.

    This influence of these factors are not fully covered in this paper.

**Reasons To Accept:**

1. the designed SSE metrics is simple yet effective, which highly correlated with evaluation on benchmarks

**Reasons To Reject:**

1. Until I finished reading the entire paper, I found it challenging to grasp the specific problem the paper aimed to address. The description of the issue in the introduction, as well as the analysis of the shortcomings of related work, was not clear.
   1. I understand the solved problem in this paper in conclusion section "The key advantage of self-sensitivity evaluation is that it has the potential to remove the need to laboriously label new data, leading to more efficient forms of evaluation in real deployment settings. We showcase several case studies".
   2. If this is the problem to be solved, an obvious issue arises. The paper lacks an effective baseline for comparison and analysis to demonstrate the superiority of the proposed self-sensitivity method. For instance, a very simple (and likely effective) approach would be to directly use LLM-as-a-judge [1] to evaluate model quality. Extensive prior work has demonstrated that powerful closed-source models like GPT-4 have a significant advantage over human evaluation in assessing the quality of language model generation. This advantage could also address the problem raised by the authors, i.e., "it has the potential to remove the need to laboriously label new data, leading to more efficient forms of evaluation in real deployment settings".

[1] Judging LLM-as-a-Judge with MT-Bench and Chatbot Arena https://arxiv.org/abs/2306.05685

---

> ### Author Rebuttal · Authors · 2024-05-31
>
> Thank you, zqFn.
>
> > On [1] to evaluate model quality.
>
> This raises a valid point regarding an LLM as a judge. However, the form of evaluation mentioned in [1] is specifically tailored for chatbots or models that follow instructions. In contrast, our self-sensitivity metrics cater to a broader class and are more general, evaluating any LLMs, including base models. For instance, similar to other evaluations found in the LM-Eval Harness, we do not judge the generations except for toxicity. In cases of toxicity, we demonstrate that our method correlates very well when using a model as the judge, specifically the PerspectiveAPI by Google. The advantages of our method compared to using a model as a judge are that it is much more *efficient*.
>
> Additionally, our method seeks to understand the model's behavior without focusing on the model's generation. We include a perplexity baseline in our Knowledge metric, which shows that perplexity does not always accurately capture model behavior. For example, Galactica, a specialized science language model, exhibits high perplexity (where higher is worse) but scores well on the MMLU clinical metric . This discrepancy is not observed in our method, which accurately predicts Galactica's performance.
>
> [1] https://arxiv.org/abs/2306.05685
>
> >Figure 2 seems to missing the performance on the "Negations".
>
> Apologies for the confusion. We utilized negations to determine closed-book knowledge. In Section 4, we detail how we used negations.
>
> > Questions regarding Bookcorpus usage
>
> We apologize for the confusion. In Section 4, "Knowledge with Negations," we show that the normalized sensitivity score correlates with labeled datasets. The unnormalized sensitivity metric uses Wikipedia as the corpus to calculate the score. We use BookCorpus to help normalize our metrics. The goal of this normalization is to control for higher sensitivity scores from Wikipedia due to the presence of the word "not" in the sentences. To achieve this, we use BookCorpus, which does not contain factual information, to control for this effect. Figure 12 (left) in App. shows the plot without normalizing for the appearance of "not" in the sentence. Here, we see that the instruction-tuned models are more dispersed compared to Figure 3 (left). Additionally, we believe it is an interesting property of our evaluation that, in general, these transformations only need to be informative on average (Appendix A.5, "Do All Transformations Need to Be Informative?").

---

> > ### Comment · Reviewer_zqFn · 2024-06-07
> >
> > Thank you for your response.
> >
> > After reading your response, I increased the score.

---

### Official Review · Reviewer_vbKk · 2024-05-11

**Rating:** 6
**Confidence:** 4
**Ethics Flag:** 1

**Summary:**

This paper introduces a new framework that does not rely on human-annotated data for LLM evaluation. Specifically, the paper’s methodology involves modifying the input through negation and various techniques, thereby evaluating the robustness of LLMs. The work incorporates several self-supervised evaluation strategies to thoroughly assess different facets of LLMs, along with detailed case studies. Notably, the experimental results demonstrate a positive correlation between the proposed metric and performance on downstream tasks, indicating its potential as a valuable complement to traditional evaluation methods.

**Questions To Authors:**

I'm curious about how many of the generated samples are meaningful.

**Reasons To Accept:**

1. This paper is well-organized and easy to follow.
2. This paper presents a new approach to the evaluation of large language models (LLMs). By eliminating the dependence on human-annotated labels, it paves a new path for more objective and scalable evaluation methods.
3. The paper’s methodology can be seamlessly applied to diverse domains and data types, which is valuable in realistic settings.

**Reasons To Reject:**

1. The key idea of modifying the input sentence and measuring sensitivity or invariance has similarities to robustness testing, which has been investigated in previous work [1]. It would be beneficial to better position the contribution of this paper.
2. The proposed input modification method is somewhat trivial, this simple method might generate meaningless samples, thus affecting the main claim of this paper. It would be better to analyze to what extent such samples affect the claim of this paper.

[1] ON THE ROBUSTNESS OF CHATGPT: AN ADVERSARIAL AND OUT-OF-DISTRIBUTION PERSPECTIVE. ICLR2023 Workshop.

---

> ### Author Rebuttal · Authors · 2024-05-31
>
> Thank you for your time, vbKk
>
> >  Position the contribution to [1]
>
> Thank you for bringing up this work. We will include it in our revised version of the paper.
>
> Regarding the differences, [1] considers adversarial examples to the original input and label pairs. We believe this aligns more with [2], which we have included in the related works. The differences themselves stem from the fact that we consider labels, whereas [1] and [2] do not require them. In a sense, our evaluation is self-supervised.
>
> To conclude, while the methods in [1] and [2] rely on existing labeled data, our method departs from these previous works as we analyze invariances using a data-agnostic, label-free procedure.
>
> [1] https://arxiv.org/pdf/2302.12095
> [2] https://aclanthology.org/2020.acl-main.442/
>
> > On the transformations
>
> Our evaluation method generally only requires transformations to be informative on average (please see Appendix A.5, "Do All Transformations Need to Be Informative?"). Even if some transformations are only informative for a specific dataset, there is still enough signal to compare relative model performance, although the scores may be lower. We view this robustness as a key strength of our approach. Furthermore, we conducted a quick human examination of many of the transformations to confirm that they were indeed meaningful.
>
> The self-sensitivity metric necessitates human design, as is the case with ALL evaluation methods. However, our self-sensitivity framework offers a more streamlined and efficient approach compared to conventional evaluation methods. While domain-specific adaptations are necessary, they are less time-consuming and complex than creating entirely new benchmarks for each specific problem. Additionally, our framework's flexibility allows it to be tailored to a wide range of domains and scale to the entire size of a corpus.

---

> > ### Comment · Reviewer_vbKk · 2024-06-04
> > **Thanks for your response**
> >
> > Your response regarding weakness #1 has addressed my concerns. As for weakness #2, could you describe how you evaluated the transformed data? Do you have any quantitative results, and could you provide some examples?

---

> > > ### Author Response · Authors · 2024-06-05
> > > **Response and thanks for the engagement**
> > >
> > > We manually examined 100 examples during development, which is significantly cheaper than constructing a dataset. In these hand-examined examples from each transformation, we found that almost all transformations were as intended, except for some instances where shifts in word order caused the misplacement of quotes or commas. For instance, in knowledge probing via negations, we applied "not" to topic sentences and found all to be perfect counterfactuals of the original sentences. However, this may not hold if the document was malformed (e.g., HTML/Tables), which was not the case for us. We also provide real examples in the Appendix. Furthermore, in Appendix A.5, we present a mathematical argument demonstrating that sensitivity scores need only a small signal (just larger than the standard error) to be effective. Additionally, it is crucial to note that our goal is to validate the SSE framework, which we examine in five different settings.
> > >
> > > We appreciate the reviewer to take the time to respond to our rebuttal.

---

> > > > ### Comment · Reviewer_vbKk · 2024-06-07
> > > > **Thanks for your response**
> > > >
> > > > My concern has been addressed, so I have increased the score accordingly.

---

### Official Review · Reviewer_NEer · 2024-05-11

**Rating:** 6
**Confidence:** 4
**Ethics Flag:** 1

**Summary:**

This paper proposes a self-sensitive evaluation paradigm that can be used to evaluate LLMs from multiple aspects, such as closed-book knowledge and toxicity.
The main idea is to transfer the original corpus into a form of transformation and measure the similarity between the original one and the transformed one.
The paper shows evaluation over multiple metrics, and compares several LLMs, such as Davinci, LLaMA, etc.

**Reasons To Accept:**

This paper is generally well written.
The proposed method is easy and can be extended to other datasets.
Most analyses are interesting.
Also, kudos to the authors as they include corresponding discussion of their limitations.

**Reasons To Reject:**

1. The overall methods are rather human-designed, i.e. each transformation of the corpora requires human designs. For example, inserting "not" into the sentences.

2. Such word-level transformation can be simple and can be easily extended. However, it can be biased and less robust in the sense that, for example, there are multiple ways of expressing "not", and models can show different performances towards such transformation.

---

> ### Author Rebuttal · Authors · 2024-05-31
>
> Thank you for your time, Reviewer NEer.
>
> > Each transformation of the corpora requires human design and can be biased and less robust in the sense that
>
> We agree that the method requires human design; however, so do all evaluations. All evaluations necessitate human design to develop the required input and label pairs. This procedure fundamentally takes much longer to construct than our self-sensitivity framework, which remains significantly more efficient than conventional forms of evaluation.
>
> Regarding the bias and robustness of the transformations, we believe it is an interesting property of our evaluation that, in general, these transformations only need to be informative on average (see Appendix A.5, "Do All Transformations Need to Be Informative?"). If some of the transformations are informative for a given corpus, then, although scores may be lower, sufficient signal exists to compare relative model performance. We actually consider this robustness a core strength of our approach and would be happy to discuss this further. That being said, we agree that the negation process could be made more general. One way to do this would be to ask another model, like ChatGPT, to construct a counterfactual to the sentence.

---

> > ### Comment · Reviewer_NEer · 2024-06-05
> > **Response to rebuttal**
> >
> > Thanks for authors' response. After reading the response, I think my score is ok and will not change it.

---

### Official Review · Reviewer_bvvQ · 2024-05-12

**Rating:** 6
**Confidence:** 4
**Ethics Flag:** 1

**Summary:**

This work proposed a new evaluation method based on comparison between a text its negations. In detail, authors proposed to make handcrafted negation perturbation on original text and compute various difference between this counterpart and its originals, like perplexity, token softmax probability distribution or text. Authors proposed this new test method can provide more accurate metrics for various domain datasets.

Authors motivation contains these points:
1. New general automatic evaluation for LLM text generation ability is an interesting topic and authors do provide a lot of interesting results.
2. Authors stressed the importance of data contamination and intended to propose a corresponding solution.
3. They hope to have this new automatic evaluation metric to be highly aligned with human intention.

**Questions To Authors:**

1. The authors mentioned a lot about data contamination in the introduction part. However, I can not understand how evaluation metric modification can migrate this.
2. How do authors compute the perplexity of API-based models like Cohere and OpenAI? How much does it cost to compute perplexity?
3. Is there any evidence to show that self-sensitivity evaluation can exhibit the difference among different pertained LLMs?

**Reasons To Accept:**

1. The new general automatic evaluation is interesting. The correlation between the proposed metric and its corresponding domain-specific metrics seems highly correlated.
2. In several dimensions, the proposed evaluation way do prove the difference between the pre-trained version and instruction-tuned version of LLMs.
3. For different problems, authors provide specific solutions to adjust their unique patterns.

**Reasons To Reject:**

1. Although the proposed metric seems like a new general evaluation way, it also requires a lot of design to get it to work on various domains like toxicity, closed-book knowledge, and word order sensitivity. Now that we still need to consider a lot domain-specific features to get self-sensitivity metric work, why not directly turn to previous evolution ways on benchmarks? How to pick the best-matched metric for self-sensitivity evaluation in specific problems remains a major issue when I plan to use it. I did not get the convenience of using self-sensitivity evaluation.
2. The authors did not provide the correlation between the proposed self-sensitivity evaluation and human evaluation results. They still only compute the correlations between the proposed self-sensitivity evaluation and the original automatic evaluation metrics.  I suggest this can not fully prove the effectiveness of the proposed evaluation ways.
3. The existing evaluation results are not complex enough. It is better to incorporate more datasets as well as new models to prove self-sensitivity evaluation’s effectiveness.
4. The writing structure is a little confusing. I did not get the reason why authors put context sensitivity and tokenization robustness in the appendix while others on the main context.
5. The author suggests the proposed evaluation is dataset-agnostic. However, I find that different evaluation datasets need different self-sensitivity function designs. Does this mean self-sensitivity evaluation is still dataset-centric?

---

> ### Author Rebuttal · Authors · 2024-05-31
>
> Thank you, bvvQ.
>
> > Convenience of SSE.
>
> We agree self-sensitivity metric necessitates human design, as with all evaluations. Every evaluation process requires significant human effort to develop appropriate input and label pairs. However, our self-sensitivity framework offers an *efficient* approach compared to labeled evaluations as there is no need to laboriously collect data. While domain-specific adaptations are necessary, they are less time-consuming and complex than creating entirely new benchmarks for each specific problem.
>
> > SSE and human evaluation
>
> Human evaluation is the gold standard for many aspects of model performance. However, due to the reproducibility issues and laborious nature of human evaluation, labeled data is still used ubiquitously. Our primary aim was to show the advantages of our SSE relative to existing automatic evaluation metrics. This approach aims to overcome several limitations associated with labeled data, such as data contamination, laborious collection, and scalability issues.
>
> > data-centric
>
> Our evaluation is dataset-agnostic, which should be understood as its flexibility and applicability to various types of data, a "one-size-fits-many" approach rather than "one-size-fits-all." SSE still necessitates tailoring to the specific characteristics of each dataset. For instance, in knowledge, the focus is on ensuring that factual content exists, irrespective of the proportion of non-factual statements, as long as the standard error remains below the sensitivity score (see Appendix A.5). A dataset with no factual sentences such as fictional books would not be an ideal candidate for the knowledge SSE metric.
>
> >  Data Contamination
>
> SSEs are meant to be data-agnostic in that a new dataset from live sources can be used to evaluate the data. For example, if MMLU clinical questions are contaminated, then the knowledge SSE can be run on new clinical articles, as the model will never have seen these samples.
>
> > API Questions
>
> API-based models sometimes provide the negative log-likelihood per token for input tokens. However, as API costs evolve, it becomes harder to predict exactly the costs.
>
> > Differences in LLMs
>
> We do see that there is a particular ordering among different models across the various metrics we considered. In almost all cases (except for tokenization), higher sensitivity scores mean that the model is better.

---

### Decision · Program_Chairs · 2024-07-10

**Decision:**

Accept

**Comment:**

This paper proposes a self-sensitivity evaluation for LLMs to evaluate their behavior on unseen data. The method proposes applying transforms to LLMs and evaluating the variation of the similarity scores. The paper studies average change in perplexity of models on perturbed sentences. After rebuttal phase, the paper has received consistent score, achieving consensus among reviewers. Two reviewers reported increased scores based on the rebuttal.The paper's strengths lie in reducing the amount of human intervention in evaluating LLMs. There are common weaknesses between the reviewers stating the lack of evaluation against human reviewers and the need for perturbation rules (such as negation). Results show correlations between sensitive scores and a number of model families, indicating that models could be evaluated with fewer human-labelled instances.